# Clinical outcomes in patients with muscle disorders and acute ischemic stroke

**Adeel S. Zubair, Lauren Littig⦿, Daniel B. DiCapua, Adam de Havenon⦿***

Department of Neurology, Yale University School of Medicine, New Haven, Connecticut

* adam.dehavenon@yale.edu

## Abstract

### Objectives

Stroke is a leading cause of morbidity in the United States, which prompts an exploration into its associated risk factors. Muscle disorders have been linked to an increased risk of stroke; however, research on stroke prevalence and clinical outcomes in this population is limited by small sample sizes. Although the diagnostic and treatment protocols for acute ischemic stroke are largely identical in patients with and without muscle disorders, it remains unclear whether clinical outcomes differ. This study aimed to address this gap using a large, nationally representative dataset.

### Materials and methods

We conducted a retrospective analysis of the 2016–2022 National Inpatient Sample to compare clinical outcomes in patients with and without muscle disorders hospitalized for an acute ischemic stroke.

### Results

Patients with pre-existing muscle disorders had significantly higher rates of intubation, PEG tube placement, and in-hospital mortality, as well as a lower likelihood of discharge to home, compared to those without muscle disorders.

### Conclusions

These findings suggest that patients with muscle disorders experience worse outcomes following an ischemic stroke. Further research is needed to identify the underlying factors driving these disparities and to inform targeted strategies for improving outcomes in this vulnerable population.

**Data availability statement:** The data are publicly available from the National Inpatient Sample database (https://hcup-us.ahrq.gov).

**Funding:** The author(s) received no specific funding for this work.

**Competing interests:** The remaining authors declare that no competing interests exist.

## Introduction

Stroke is one of the leading causes of long-term disability and mortality in the United States. [1–3] While numerous medical conditions are associated with an increased risk of stroke, [4] neuromuscular disorders, such as muscular dystrophies, mitochondrial disorders, and myopathies, are also known to elevate stroke risk, yet this relationship has received comparatively little attention. [5–7]

The etiologies of stroke in patients with muscle disorders generally mirror those in the general population and include cardioembolic events, artery-to-artery embolism, and small-vessel disease. However, certain subtypes, such as dermatomyositis and polymyositis, are associated with autoimmune-driven hypercoagulability and elevated stroke risk through conditions like antiphospholipid syndrome. [7] Despite these potential differences in pathophysiology, the clinical workup and treatment for acute ischemic stroke is identical in patients with and without muscle disorders. [4–7]

Research on stroke prevalence and outcomes in patients with muscle disorders remains limited and is primarily drawn from small case studies or retrospective cohort studies. For example, one study reported a stroke prevalence of 1.5% in a cohort of patients with myopathies, similar to the general population [5–7], however the small sample size (n = 131) was a major limitation. Certain muscle disorders, such as Duchenne muscular dystrophy, have been more frequently associated with stroke [9], yet another study reported a relatively low incidence (0.75%) among 665 individuals with the condition. [7–10] These inconsistencies highlight the need for broader, population-based studies to clarify the relationship between muscle disorders and stroke outcomes.

To address this gap, the present study leverages a large, nationally representative database to evaluate clinical outcomes among patients with and without muscle disorders hospitalized for an acute ischemic stroke.

## Materials and methods

This study utilized the 2016–2022 National Inpatient Sample (NIS) database, the largest publicly available inpatient healthcare database in the United States, to conduct a retrospective analysis of non-elective hospital admissions for adult patients (age ≥ 18 years) with a primary discharge diagnosis of ischemic stroke (ICD-10-CM I63.x). Patients with missing demographic data were excluded as were those with a missing NIH Stroke Scale (NIHSS) score. NIHSS (ICD-10-CM R29.7x) was first included in the NIS database in 2016, and in this study represents the initial NIHSS upon admission. [11] Although limiting the cohort to patients with available NIHSS scores could introduce selection bias, the inclusion of stroke severity was prioritized due to its importance in modeling outcomes. [12] This study used publicly available, de-identified data and was thus exempt from review by the Yale University Institutional Review Board. Informed consent was not obtained by the Centers for Medicare & Medicaid Services (CMS) for the NIS because the dataset does not contain identifiable information.

The primary exposure was the presence of a muscle disorder, defined as the ICD-10-CM G71-72.x in any secondary diagnosis field. The ICD-10 codes G71-72 include conditions such as Duchenne muscular dystrophy, Emery-Dreifuss muscular dystrophy, facioscapulohumeral muscular dystrophy, limb-girdle muscular dystrophy, muscular dystrophy, myopathy, inflammatory and immune myopathies, alcoholic myopathy, and drug-induced myopathy.

The primary outcomes were in-hospital mortality and favorable discharge, defined as discharge to home or self-care. Secondary outcomes included stroke severity (measured by NIHSS score), stroke interventions (intravenous thrombolytic or endovascular thrombectomy), and in-hospital complications indicative of poor outcomes (intubation or PEG tube placement). Definitions of the exposures, outcomes and their adjudication are detailed in S1 Table.

Descriptive statistics were reported to compare ischemic stroke patients with and without muscle disorders. To derive odds ratios (ORs) for the outcomes, we used a multivariable logistic regression model *a priori* adjusted for the covariates of patient age (<55, 55–64, 65–74, ≥ 75), sex, comorbidities (hypertension, diabetes, atrial fibrillation, congestive heart failure, obesity), stroke interventions (endovascular thrombectomy or intravenous thrombolysis, except models fit to that), admission NIHSS score, patient urban-rural residence, primary payer, income quartile by zip code, hospital Census region, teaching status, and hospital bed size (small, medium, large), determined by the NIS using census region, hospital urban-rural designation, and teaching status.

Marginal effects were calculated to estimate the predicted probability of our outcomes for change in specific variables while holding other variables at their average predicted value after logistic regression. Variance inflation factor of each model was calculated to ensure acceptable multicollinearity, defined as a mean value of <5 and <5 for each covariate. [13] We also confirmed goodness-of-fit with the Hosmer-Lemeshow test. [14] All statistical analysis was performed in Stata 17.0 (StatCorp, College Station, TX).

## Results

The NIS included 742,803 patients hospitalized for an acute ischemic stroke between 2016 and 2022. After excluding 395,889 patients with missing NIHSS scores and 11,443 with missing covariate data, a total of 335,100 patients were included in our analysis, of whom 371 (0.11%) had a previously diagnosed muscle disorder. In the full cohort, 48.6% of patients were female, 67.9% were White, and the mean age was 69.6 years. Table 1 shows the baseline characteristics of patients stratified by the presence or absence of a muscle disorder diagnosis.

Patients with muscle disorders had a lower prevalence of hypertension (79.8% vs. 86.2%, p<0.001), yet were more likely to have congestive heart failure (25.6% vs. 18.0%, p<0.001), a longer hospital stay (mean of 7.9 days vs. 5.0 days, p<0.001), higher NIHSS scores (mean of 8.1 vs. 6.6, p<0.001) (Fig 1), and a higher risk of in-hospital mortality (7.5% vs. 3.4%, p<0.001), intubation (10.2% vs. 3.7%, p<0.001), and PEG tube placement (6.5% vs. 2.9%, p<0.001).

After adjusting for demographic, clinical, and hospital-level factors in a multivariate analysis, patients with muscle disorders had more than twice the odds of intubation (OR 2.42, 95% CI 1.66–3.54) and in-hospital mortality (OR 2.03, 95% CI 1.32–3.10), were significantly more likely to have a PEG tube placed (OR 1.85, 95% CI 1.19–2.86) and were significantly less likely to be discharged to home or self-care (OR 0.52, 95% CI 0.39–0.68). There were no significant differences in the use of acute stroke interventions, including intravenous thrombolysis or endovascular thrombectomy, between the two groups (Table 2).

## Discussion

This study demonstrates that patients with pre-existing muscle disorders experience worse outcomes when hospitalized for an ischemic stroke, including higher rates of intubation, PEG tube placement, and in-hospital mortality, as well as a lower likelihood of discharge to home.

Notably, patients with muscle disorders had a lower prevalence of hypertension, one of the most significant stroke risk factors, yet experienced a higher prevalence of stroke. This suggests that underlying myopathies may confer an

**Table 1. Weighted patient demographics, acute stroke interventions, and hospital characteristics for participants in our cohort with and without muscle disorders.**

| Variable | Cohort without Muscle Disorders (n = 335,100) | Cohort with Muscle Disorders (n = 371) | P value |
|---|---|---|---|
| Age, y | | | 0.182 |
| <55 | 48,797 (14.6%) | 65 (17.5%) | |
| 55-64 | 68,331 (20.4%) | 71 (19.1%) | |
| 65-74 | 85,136 (25.4%) | 103 (27.8%) | |
| ≥75 | 132,836 (39.6%) | 132 (35.6%) | |
| Male sex | 172,107 (51.4%) | 197 (53.1%) | 0.503 |
| Race/Ethnicity | | | 0.633 |
| White | 227,820 (68.0%) | 256 (69.0%) | |
| Black | 58,298 (17.4%) | 54 (14.6%) | |
| Hispanic | 27,859 (8.3%) | 35 (9.4%) | |
| Asian or Pacific Islander | 10,575 (3.2%) | 15 (4.0%) | |
| Native American | 1,496 (0.4%) | 2 (0.5%) | |
| Other | 9,052 (2.7%) | 9 (2.4%) | |
| Hypertension | 288,701 (86.2%) | 296 (79.8%) | <0.001 |
| Diabetes | 128,428 (38.3%) | 143 (38.5%) | 0.931 |
| CHF (Congestive Heart Failure) | 60,478 (18.0%) | 95 (25.6%) | <0.001 |
| Atrial Fibrillation | 84,573 (25.2%) | 103 (27.8%) | 0.263 |
| Obesity | 56,138 (16.8%) | 56 (15.1%) | 0.393 |
| Hospital Census Region | | | 0.460 |
| Northeast | 57,181 (17.1%) | 60 (16.2%) | |
| Midwest | 76,921 (23.0%) | 98 (26.4%) | |
| South | 143,249 (42.7%) | 150 (40.4%) | |
| West | 57,749 (17.2%) | 63 (17.0%) | |
| Patient Location | | | 0.181 |
| Large metro w/ ≥ 1 million | 97,853 (29.2%) | 103 (27.8%) | |
| Suburb of large metro | 82,742 (24.7%) | 109 (29.4%) | |
| Metro with 250-999k | 72,960 (21.8%) | 70 (18.9%) | |
| Metro with 50-249k | 30,233 (9.0%) | 40 (10.8%) | |
| Micropolitan | 29,540 (8.8%) | 30 (8.1%) | |
| Rural | 21,772 (6.5%) | 19 (5.1%) | |
| Expected Primary Payer | | | 0.091 |
| Medicare | 210,637 (62.9%) | 248 (66.8%) | |
| Medicaid | 32,635 (9.7%) | 38 (10.2%) | |
| Private Insurance | 67,504 (20.1%) | 68 (18.3%) | |
| Self-pay | 14,480 (4.3%) | 5 (1.3%) | |
| No Charge | 1,170 (0.3%) | 1 (0.3%) | |
| Other | 8,674 (2.6%) | 11 (3.0%) | |
| Teaching Hospital | 265,791 (79.3%) | 300 (80.9%) | 0.463 |
| Hospital bed size | | | 0.784 |
| Large | 188,817 (56.3%) | 211 (56.9%) | |
| Medium | 91,320 (27.3%) | 104 (28.0%) | |
| Small | 54,963 (16.4%) | 56 (15.1%) | |

*(Continued)*

**Table 1.** (Continued)

| Variable | Cohort without Muscle Disorders (n = 335,100) | Cohort with Muscle Disorders (n = 371) | P value |
|---|---|---|---|
| Median household income in patient's ZIP code | | | 0.212 |
| >$64,000 | 66,094 (19.9%) | 85 (23.0%) | |
| $48,000–$63,999 | 80,574 (24.3%) | 93 (25.2%) | |
| $38,000–$47,999 | 86,078 (26.0%) | 98 (26.6%) | |
| <$37,999 | 98,612 (29.8%) | 93 (25.2%) | |
| Length of hospital stay ≥10 Days | 30,193 (9.0%) | 76 (20.5%) | <0.001 |
| Length of stay (continuous) | 5.0 (6.3) | 7.9 (9.9) | <0.001 |
| NIHSS | 6.6 (7.2) | 8.1 (8.2) | <0.001 |
| EVT | 26,911 (8.0%) | 40 (10.8%) | 0.051 |
| tPA | 44,590 (13.3%) | 41 (11.1%) | 0.201 |
| Died during hospitalization | 11,534 (3.4%) | 28 (7.5%) | <0.001 |
| Good Outcome | 124,773 (37.2%) | 85 (22.9%) | <0.001 |
| Intubation | 12,392 (3.7%) | 38 (10.2%) | <0.001 |
| PEG tube placement | 9,780 (2.9%) | 24 (6.5%) | <0.001 |

independent stroke risk or influence stroke severity and outcomes. Our finding that patients with muscle disorders experienced higher rates of congestive heart failure may help explain this apparent paradox, as cardiac muscle is similarly susceptible to the pathophysiologic impacts of myopathies.

Identifying and treating stroke in this population can be particularly challenging, as baseline weakness from their underlying condition may obscure stroke symptoms and lead to delays in diagnosis and care. Prior research has shown that patients presenting with non-traditional stroke symptoms are at risk of being misdiagnosed, emphasizing the importance of heightened clinical vigilance in this group. [15]

Certain muscle disorders have been associated with an increased risk of stroke, [7] however, no large-scale studies to date have examined stroke outcomes in patients with muscle disorders at a granular level. Our findings highlight the need to consider pre-existing myopathies in stroke prognosis and management.

## Limitations

This study is limited by the NIS's lack of data on key clinical variables, such as time from stroke onset to presentation, prior functional status, and contraindications to ischemic stroke treatments, all of which may influence interventions and outcomes and confound the observed associations. Additionally, the use of administrative datasets introduces the potential for misclassification bias, and the relatively small number of ischemic stroke patients with muscle disorders limits the study's statistical power. This also limited our ability to stratify by specific type of myopathy, which likely carry different risks for stroke and related complications. Future studies using disease-specific registries or prospective designs are needed to identify subgroups that are the true drivers of increased risk. Lastly, survey design adjustments were not performed to prevent introducing bias for a rare exposure. Despite these limitations, this study is strengthened by the use of a large, nationally representative dataset encompassing all-payer claims and the most up-to-date information available.

## Conclusion

Patients with pre-existing muscle disorders hospitalized for an acute ischemic stroke experience higher rates of adverse outcomes, including intubation, PEG tube placement, in-hospital mortality, and a lower likelihood of being discharged to home. Further research is needed to understand the drivers of these disparities and to develop targeted strategies for improve clinical management and outcomes in this vulnerable population.

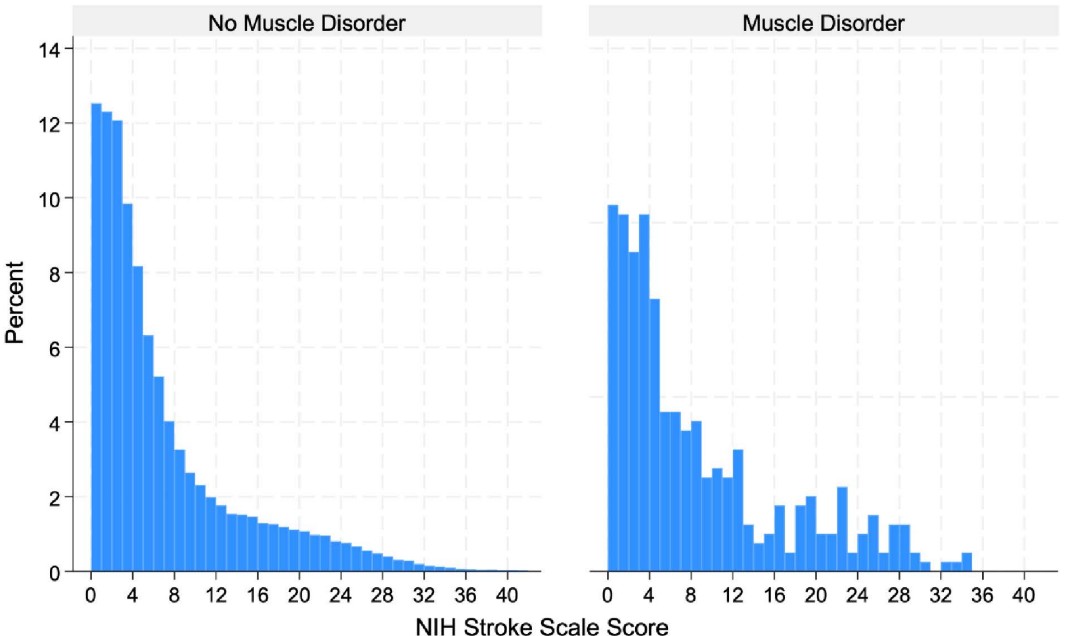

**Fig 1. Distribution of NIH Stroke Scale Scores by Presence or Absence of a Muscle Disorder Diagnosis.**

**Table 2. Odds ratios for in-hospital death, discharge home, intravenous thrombolysis, endovascular thrombectomy, intubation, and percutaneous endoscopic gastrostomy (PEG), shown for individuals with myopathic disease.**

| Outcome | Odds Ratio* | 95% CI | p value** |
|---|---|---|---|
| In-hospital death | 2.03 | 1.32-3.10 | 0.001 |
| Discharge home | 0.52 | 0.39-0.68 | <0.001 |
| Thrombolysis | 0.72 | 0.52-1.01 | 0.054 |
| Thrombectomy | 1.06 | 0.73-1.53 | 0.775 |
| Intubation | 2.42 | 1.66-3.54 | <0.001 |
| PEG | 1.85 | 1.19-2.86 | 0.006 |

*Adjusted for patient age (<55, 55–64, 65–74, ≥ 75), sex, hypertension, diabetes, atrial fibrillation, congestive heart failure, obesity, endovascular thrombectomy (except model fit to that), intravenous thrombolysis (except model fit to that), patient urban-rural residence, primary payor, quartile of income by zip code, hospital Census region, teaching status, bed size (small, medium, large), and admission NIH Stroke Scale.

## Supporting information

**S1 Table. ICD-10 codes and NIS data used for definitions of variables.** (DOCX)

## Author contributions

**Conceptualization:** Adeel S. Zubair, Daniel B. DiCapua, Adam de Havenon.

**Data curation:** Adeel S. Zubair, Daniel B. DiCapua.

**Formal analysis:** Adeel S. Zubair.

**Project administration:** Lauren Littig.

**Writing – original draft:** Lauren Littig.

**Writing – review & editing:** Lauren Littig.

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
