## [Editor Report · Decision Letter 0]

27 Jan 2025

PONE-D-24-54767Clinical outcomes in patients with muscle disorders and acute ischemic strokePLOS ONE

Dear Dr. de Havenon,

Thank you for submitting your manuscript to PLOS ONE. After careful consideration, we feel that it has merit but does not fully meet PLOS ONE’s publication criteria as it currently stands. Therefore, we invite you to submit a revised version of the manuscript that addresses the points raised during the review process.

This is an interesting study.

There is novelty in this research. 

I highlight some methodological flaws in design. 

I give some comments and suggestions. 

- Please be detail in the method. Is it retrospective, cross sectional, or cohort ? Cross sectional is smash-spot study. In table, the authors use cohort. Please be consistent. 

- The method should highlight the definition of muscle disease, the type of muscle disease, the specific treatment. 

- The discussion should begin with the findings of your study. 

- The discussion should elaborate more the findings of the study. 

- The discussion should highlight more the mechanism and biological plausibility of this findings. 

- The conclusion should answer the clinical question. The conclusion should be concise and clear.

We look forward to receiving your revised manuscript.

Kind regards,

Rizaldy Taslim Pinzon

Academic Editor

PLOS ONE

Journal Requirements:

“DBD has served on the speaker’s bureau and as a consultant for argenx and reports no competing interests related to this work. The remaining authors declare that no competing interests exist.”

3. Please note that your Data Availability Statement is currently missing the repository name and/or the DOI/accession number of each dataset OR a direct link to access each database. If your manuscript is accepted for publication, you will be asked to provide these details on a very short timeline. We therefore suggest that you provide this information now, though we will not hold up the peer review process if you are unable.

Additional Editor Comments:

This is an interesting study.

There is novelty in this research.

I highlight some methodological flaws in design.

I give some comments and suggestions.

- Please be detail in the method. Is it retrospective, cross sectional, or cohort ? Cross sectional is smash-spot study. In table, the authors use cohort. Please be consistent.

- The method should highlight the definition of muscle disease, the type of muscle disease, the specific treatment.

- The discussion should begin with the findings of your study.

- The discussion should elaborate more the findings of the study.

- The discussion should highlight more the mechanism and biological plausibility of this findings.

- The conclusion should answer the clinical question. The conclusion should be concise and clear.

---

## [Author Response · Author response to Decision Letter 1]

6 Mar 2025

March 5, 2025

We would like to thank the Editorial Board and Reviewers for the opportunity to submit a revised version of “Clinical outcomes in patients with muscle disorders and acute ischemic stroke” to PLOS ONE. We have responded to the comments below.

Reviewer #1: This is an interesting study. There is novelty in this research. I highlight some methodological flaws in design. I give some comments and suggestions:

1. Please be detail in the method. Is it retrospective, cross sectional, or cohort? Cross sectional is smash-spot study. In table, the authors use cohort. Please be consistent.

Our study is retrospective, and we have revised the manuscript in all relevant places to state this. We keep use of the word “cohort” to describe our study population, such as in places like the table.

2. The method should highlight the definition of muscle disease, the type of muscle disease, the specific treatment.

We already included which muscle disorders were our primary exposures in the methods section and listed their corresponding ICD-10 codes (G71-72). We did not detail specific treatments for each condition in the methods section as we felt this would distract from the focus of our study and analysis. However, we have included a short paragraph describing the treatment of muscle disorders in the introduction as background information.

3. The discussion should begin with the findings of your study. The discussion should elaborate more the findings of the study.

Thank you for the comment. We have reiterated findings more explicitly.

4. The discussion should highlight more the mechanism and biological plausibility of this findings.

We have added in that the mechanism of these worse outcomes can be potentially secondary to patient baseline weakness and poor functional reserve, leading to less likelihood of being discharged home and having increased risk of mortality.

5. The conclusion should answer the clinical question. The conclusion should be concise and clear.

We have added a conclusion section with a succinct and clear line about the findings and answering our clinical question about outcomes of these patients.

---

## [Decision Letter · Decision Letter 1]

28 May 2025

PONE-D-24-54767R1Clinical outcomes in patients with muscle disorders and acute ischemic strokePLOS ONE

Dear Dr. de Havenon,

Thank you for submitting your manuscript to PLOS ONE. After careful consideration, we feel that it has merit but does not fully meet PLOS ONE’s publication criteria as it currently stands. Therefore, we invite you to submit a revised version of the manuscript that addresses the points raised during the review process.

We look forward to receiving your revised manuscript.

Kind regards,

Atakan Orscelik

Academic Editor

PLOS ONE

Reviewers' comments:

Reviewer's Responses to Questions

**Comments to the Author**

1. If the authors have adequately addressed your comments raised in a previous round of review and you feel that this manuscript is now acceptable for publication, you may indicate that here to bypass the “Comments to the Author” section, enter your conflict of interest statement in the “Confidential to Editor” section, and submit your "Accept" recommendation.

Reviewer #1: All comments have been addressed

Reviewer #2: All comments have been addressed

Reviewer #3: (No Response)

Reviewer #4: (No Response)

2. Is the manuscript technically sound, and do the data support the conclusions?

Reviewer #1: No

Reviewer #2: Yes

Reviewer #3: Partly

Reviewer #4: Yes

3. Has the statistical analysis been performed appropriately and rigorously? 

Reviewer #1: Yes

Reviewer #2: Yes

Reviewer #3: Yes

Reviewer #4: Yes

4. Have the authors made all data underlying the findings in their manuscript fully available?

Reviewer #1: Yes

Reviewer #2: Yes

Reviewer #3: Yes

Reviewer #4: Yes

5. Is the manuscript presented in an intelligible fashion and written in standard English?

Reviewer #1: No

Reviewer #2: Yes

Reviewer #3: Yes

Reviewer #4: Yes

6. Review Comments to the Author

Reviewer #1: The current study does not add anything new to the current Literature.

I don't think it can be interesting.

Reviewer #2: I have no further comments. Authors revised article properly....................................................

Reviewer #3: The most significant limitation of this study is that all the different types of myopathy are combined as if they all carry similar risk for stroke related complications when we know that this is not the case. These diseases are highly variable in severity, complications, and life expectancy. Some affect swallowing function (which could require intubation and tube feeding more frequently), while others do not. To combine these subpopulations results in overestimating risk for some members of the population while underestimating risk for others. A more disease-specific approach would be more informative, as it would identify specific subgroups that are the true drivers of increased risk (to which management can then be tailored).

The authors report collecting the NIH Stroke Scale on admission and excluding patients who did not have the NIHSS. However, this data is not reported in the results. It would potentially be useful to have this information, as it would give us an idea of the distribution of stroke severity in the groups being studied.

For this type of medical record analysis, it would be useful to describe the numbers of patients that were excluded based on pre-specified eligibility criteria like the absence of the NIH Stroke Scale, as this gives us important information regarding potential selection bias.

It is more common to refer to “Duchenne muscle dystrophy” as “Duchenne muscular dystrophy.” The non-eponymous diseases, like limb-girdle muscular dystrophy and facioscapulohumeral muscular dystrophy, are usually not capitalized.

The authors report that patients with muscle disorders have a higher prevalence of hypertension, obesity, and diabetes. However, this risk difference is only statistically significant for hypertension. I would avoid grouping these risk factor differences as if they all met statistical significance. Similarly, I would avoid implying that the odds of requiring intubation or PEG tube placement were higher for those with muscle disorders when these differences did not reach statistical significance. The fact that the differences did not reach statistical significance after adjustment for potential confounders would mean that the risk is essentially the same for many readers.

In Table 1, how was the length of hospital stay converted into a binary outcome?

Discussion: “Interestingly, ischemic stroke patients with muscle disorders were more likely to have congestive heart failure which contrasts with their lower prevalence of other vascular risk factors. This paradox suggests that certain muscle disorders may contribute to cardiac dysfunction or share pathophysiologic mechanisms with heart disease, both of which warrant further investigation.” I don’t find that there is anything really paradoxical about this observation. Patients with genetic muscle diseases, which comprise most of the G71 ICD10 category, are known to develop cardiomyopathies that occur early in life before other vascular risk factors result in complications like stroke. To imply that this is a mystery that needs to be investigated seems to ignore the existence of a well-established and extensively studied medical condition. It also underestimates the readers, most of whom will know that the heart is made of muscle and can be susceptible to the effects of muscle disease.

Reviewer #4: This is an important foray into an understudied area of stroke risks and outcomes in myopathies.

Unclear what is meant by "...except models fit to that" in lines 101-102 of the Methods.

Hospital size by number of beds may be more precise than "hospital bed size" throughout the text and tables (sounds like it is referring to the size of the beds themselves). What are the cutoffs for each category?

Was there information on smoking and alcohol use to compare between the groups at baseline?

Suggest that paragraph in the Discussion starting with line 157 be revised. Both cardiomyopathy and rhythm disorders are well-established aspects of the phenotype of many myopathies - this does not seem to require further investigation in and of itself but perhaps there are ranges of EF/etc. that might increase stroke risk that are different that CHF from other etiologies.

Another limitation is that some myopathy patients may not live independently at baseline and thus the post-stroke status may not be a change for them.

This study will be a useful impetus for further investigation into this association beyond incidence epidemiology.

7. PLOS authors have the option to publish the peer review history of their article (what does this mean? ). If published, this will include your full peer review and any attached files.

**Do you want your identity to be public for this peer review?** For information about this choice, including consent withdrawal, please see our Privacy Policy .

Reviewer #1: No

Reviewer #2: No

Reviewer #3: No

Reviewer #4: No

---

## [Author Response · Author response to Decision Letter 2]

8 Jul 2025

Reviewer #1:

The current study does not add anything new to the current Literature. I don't think it can be interesting.

We respectfully disagree. To our knowledge, this is the first study using a nationally representative dataset to examine ischemic stroke outcomes in patients with muscle disorders. We have expanded the dataset to include NIS 2021–2022 data, which increases our sample size to 335,471, including 371 patients with muscle disorders, and enhances statistical power. We hope this revised version provides a clearer demonstration of the study’s contributions.

Reviewer #2:

I have no further comments. Authors revised article properly.

Thank you!

Reviewer #3:

The most significant limitation of this study is that all the different types of myopathy are combined as if they all carry similar risk for stroke related complications when we know that this is not the case. These diseases are highly variable in severity, complications, and life expectancy. Some affect swallowing function (which could require intubation and tube feeding more frequently), while others do not. To combine these subpopulations results in overestimating risk for some members of the population while underestimating risk for others. A more disease-specific approach would be more informative, as it would identify specific subgroups that are the true drivers of increased risk (to which management can then be tailored).

We agree that combining all different types of myopathies is a limitation, however stratifying by type of myopathy resulted in sample sizes too small for meaningful statistical comparisons.

While one alternative would be to stratify by ICD-10-CM codes G71/G72 (Primary disorders of muscles vs. other and unspecified myopathies), we felt this oversimplified the variation in severity, complications, and prognosis across subtypes and would not meaningfully identify the subgroups driving increased risk. We acknowledge this limitation in the discussion.

The authors report collecting the NIH Stroke Scale on admission and excluding patients who did not have the NIHSS. However, this data is not reported in the results. It would potentially be useful to have this information, as it would give us an idea of the distribution of stroke severity in the groups being studied.

Thank you, we now include NIHSS data in the Results section and in Table 1.

For this type of medical record analysis, it would be useful to describe the numbers of patients that were excluded based on pre-specified eligibility criteria like the absence of the NIH Stroke Scale, as this gives us important information regarding potential selection bias.

We agree and have added these exclusion counts to the Methods section.

It is more common to refer to “Duchenne muscle dystrophy” as “Duchenne muscular dystrophy.” The non-eponymous diseases, like limb-girdle muscular dystrophy and facioscapulohumeral muscular dystrophy, are usually not capitalized.

Thank you, we have corrected the terminology and capitalizations.

The authors report that patients with muscle disorders have a higher prevalence of hypertension, obesity, and diabetes. However, this risk difference is only statistically significant for hypertension. I would avoid grouping these risk factor differences as if they all met statistical significance. Similarly, I would avoid implying that the odds of requiring intubation or PEG tube placement were higher for those with muscle disorders when these differences did not reach statistical significance. The fact that the differences did not reach statistical significance after adjustment for potential confounders would mean that the risk is essentially the same for many readers.

Thank you. We have revised the Results section to include only statistically significant differences and clarified the interpretation of adjusted models.

In Table 1, how was the length of hospital stay converted into a binary outcome?

Hospital length of stay was dichotomized as <10 days or ≥ 10 days. We have clarified this in the manuscript!

Discussion: “Interestingly, ischemic stroke patients with muscle disorders were more likely to have congestive heart failure which contrasts with their lower prevalence of other vascular risk factors. This paradox suggests that certain muscle disorders may contribute to cardiac dysfunction or share pathophysiologic mechanisms with heart disease, both of which warrant further investigation.” I don’t find that there is anything really paradoxical about this observation. Patients with genetic muscle diseases, which comprise most of the G71 ICD10 category, are known to develop cardiomyopathies that occur early in life before other vascular risk factors result in complications like stroke. To imply that this is a mystery that needs to be investigated seems to ignore the existence of a well-established and extensively studied medical condition. It also underestimates the readers, most of whom will know that the heart is made of muscle and can be susceptible to the effects of muscle disease.

Thank you. We originally used the term “paradox” to describe the coexistence of lower hypertension prevalence yet higher CHF prevalence in patients with muscle disorders, despite both being traditional risk factors for stroke. However, as you rightly note, it shouldn’t be surprising considering the heart is susceptible to effects of muscle disease. We have revised this paragraph and use your rationale for the observation.

Reviewer #4:

This is an important foray into an understudied area of stroke risks and outcomes in myopathies.

Thank you for your encouraging feedback!

Unclear what is meant by "...except models fit to that" in lines 101-102 of the Methods.

This means that particular variable was excluded as a covariate (not adjusted for) in the model where it was the outcome.

Hospital size by number of beds may be more precise than "hospital bed size" throughout the text and tables (sounds like it is referring to the size of the beds themselves). What are the cutoffs for each category?

The cutoffs for each category of bed size are determined by the NIS using census region, hospital urban-rural designation, and teaching status. We have added this language to the methods section.

Was there information on smoking and alcohol use to compare between the groups at baseline?

Unfortunately, the NIS does not collect information on smoking status.

Suggest that paragraph in the Discussion starting with line 157 be revised. Both cardiomyopathy and rhythm disorders are well-established aspects of the phenotype of many myopathies - this does not seem to require further investigation in and of itself but perhaps there are ranges of EF/etc. that might increase stroke risk that are different that CHF from other etiologies.

Thank you, a similar comment was made by Reviewer #3 and we have revised that paragraph.

Another limitation is that some myopathy patients may not live independently at baseline and thus the post-stroke status may not be a change for them.

Great point, we have added this to our limitations section.

This study will be a useful impetus for further investigation into this association beyond incidence epidemiology.

Thank you!

---

## [Decision Letter · Decision Letter 2]

21 Jul 2025

Clinical outcomes in patients with muscle disorders and acute ischemic stroke

PONE-D-24-54767R2

Dear Dr. Havenon,

We’re pleased to inform you that your manuscript has been judged scientifically suitable for publication and will be formally accepted for publication once it meets all outstanding technical requirements.

Kind regards,

Atakan Orscelik

Academic Editor

PLOS ONE

Additional Editor Comments (optional):

Reviewers' comments:

Reviewer's Responses to Questions

**Comments to the Author**

1. If the authors have adequately addressed your comments raised in a previous round of review and you feel that this manuscript is now acceptable for publication, you may indicate that here to bypass the “Comments to the Author” section, enter your conflict of interest statement in the “Confidential to Editor” section, and submit your "Accept" recommendation.

Reviewer #4: All comments have been addressed

2. Is the manuscript technically sound, and do the data support the conclusions?

Reviewer #4: (No Response)

3. Has the statistical analysis been performed appropriately and rigorously? 

Reviewer #4: (No Response)

4. Have the authors made all data underlying the findings in their manuscript fully available?

Reviewer #4: (No Response)

5. Is the manuscript presented in an intelligible fashion and written in standard English?

Reviewer #4: (No Response)

6. Review Comments to the Author

Reviewer #4: (No Response)

7. PLOS authors have the option to publish the peer review history of their article (what does this mean? ). If published, this will include your full peer review and any attached files.

**Do you want your identity to be public for this peer review?** For information about this choice, including consent withdrawal, please see our Privacy Policy .

Reviewer #4: No

---

## [Editor Report · Acceptance letter]

PONE-D-24-54767R2

PLOS ONE

Dear Dr. de Havenon,

I'm pleased to inform you that your manuscript has been deemed suitable for publication in PLOS ONE. Congratulations! Your manuscript is now being handed over to our production team.

Kind regards,

on behalf of

Dr. Atakan Orscelik

Academic Editor

PLOS ONE